# Correlative Light-Environmental Scanning Electron Microscopy of Plasma Membrane Efflux Carriers of Plant Hormone Auxin

**DOI:** 10.3390/biom11101407

**Published:** 2021-09-26

**Authors:** Ayoub Stelate, Eva Tihlaříková, Kateřina Schwarzerová, Vilém Neděla, Jan Petrášek

**Affiliations:** 1Department of Experimental Plant Biology, Faculty of Science, Charles University, Viničná 5, 128 44 Prague 2, Czech Republic; stelatea@natur.cuni.cz (A.S.); schwarze@natur.cuni.cz (K.S.); 2Institute of Scientific Instruments, Academy of Sciences of the Czech Republic, Královopolská 147, 612 64 Brno, Czech Republic; tihlarik@isibrno.cz (E.T.); vilem@ISIBrno.cz (V.N.)

**Keywords:** correlative microscopy, plasma membrane, nanodomains, auxin carriers

## Abstract

Fluorescence light microscopy provided convincing evidence for the domain organization of plant plasma membrane (PM) proteins. Both peripheral and integral PM proteins show an inhomogeneous distribution within the PM. However, the size of PM nanodomains and protein clusters is too small to accurately determine their dimensions and nano-organization using routine confocal fluorescence microscopy and super-resolution methods. To overcome this limitation, we have developed a novel correlative light electron microscopy method (CLEM) using total internal reflection fluorescence microscopy (TIRFM) and advanced environmental scanning electron microscopy (A-ESEM). Using this technique, we determined the number of auxin efflux carriers from the PINFORMED (PIN) family (*Nt*PIN3b-GFP) within PM nanodomains of tobacco cell PM ghosts. Protoplasts were attached to coverslips and immunostained with anti-GFP primary antibody and secondary antibody conjugated to fluorochrome and gold nanoparticles. After imaging the nanodomains within the PM with TIRFM, the samples were imaged with A-ESEM without further processing, and quantification of the average number of molecules within the nanodomain was performed. Without requiring any post-fixation and coating procedures, this method allows to study details of the organization of auxin carriers and other plant PM proteins.

## 1. Introduction

Plant PM is a dynamic structure composed of lipids and proteins that forms a boundary between intracellular and cell wall space. Correct functioning of PM secures a wide spectrum of transport, signaling and metabolism processes. A number of these biological roles depend on the existence of protein complexes spatially and functionally restricted within the phospholipid matrix of PM [1]. Although PM is a highly dynamic structure, it contains functionally and structurally defined domains [2]. Based on their size, these domains are classified as nanodomains with sizes below 1 µm and microdomains with sizes above 1 µm [3,4,5]. Domain organization of a spectrum of plant PM lipids [6] and proteins is maintained by the interplay of intra- and extra-cellular mechanisms [1]. Domain organization of the PM allows correct progression of transport and signaling processes, which coordinate plant development and their reactions to abiotic and biotic stimuli from the environment [7,8]. Auxin efflux carriers from PIN family are one of the most extensively studied plant-integral PM proteins with numerous developmental roles [9]. Their heterogeneity in various polar PM domains is collectively supported by several mechanisms, including targeted secretion, constitutive recycling, and lateral diffusion within the PM [10,11].

The advent of modern super-resolution light microscopy techniques [12,13] clearly showed the localization and dynamics of numerous peripheral and integral PM proteins and lipids within specific nanodomains [14,15,16,17]. However, thanks to the lack of suitable plant-optimized CLEM protocols, it is still hard to correlate these results with the immuno-electron microscopy approach. For selectively imaging structures at the cell surface, one of the most frequently used methods is TIRFM, which is based on the detection of light reflected from the surface. In plants, this method is often limited by the existence of a cell wall and light refraction-based variable angle epifluorescence microscopy (VAEM/TIRF) is even more frequent. However, with the appropriate sample and microscope settings, TIRFM is used for visualizing processes at the plant cell periphery, in close vicinity to or within PM [18,19]. The advantage of TIRFM is that it could be performed both in vivo and in the fixed samples. Moreover, images might be post-processed by mathematical algorithms to improve the spatial resolution, such as super-resolution optical fluctuation (SOFI) [20] or super-resolution radial fluctuation (SRRF) [21].

The CLEM approach is still not frequently used in plant research, mainly because of the technically demanding fixation of highly turgescent plant cells containing large vacuole and cell wall. There are only a few protocols described recently, based on the high-pressure fixation and adaptation of the Tokuyasu approach [22]. The combination of super-resolution fluorescence and transmission electron microscopy techniques in cryo-fixed samples are also quite rare [23,24,25,26]. All these approaches require post-fixation and metal surface coating for the electron microscopy.

Here, we report on the optimization of the protocol allowing to observe fluoronanogold immunostained molecules of tobacco auxin carrier *Nt*PIN3b-GFP within the PM by TIRFM and correlate their positions with images obtained by electron microscopy. To overcome the need for any post-fixation and metal coating steps, we used A-ESEM, a new generation of ESEM, which has been developed for the observation of native-like samples at high resolution, under a low electron dose and conditions minimizing the damage of the sample [27,28]. Using this CLEM method, here, we estimate the number of *Nt*PIN3b-GFP molecules within PM nanodomains.

## 2. Materials and Methods

### 2.1. Plant Material and Culture Conditions

Cells of tobacco cell line BY-2, *Nicotiana tabacum* L. cv. Bright Yellow 2 [29], were cultured in the dark at 27 °C with constant shaking (150 rpm; orbital diameter 30 mm) and subcultured every seven days. Liquid Murashige and Skoog (MS) medium (pH 5.8) containing 3% sucrose, 4.3 g/L MS salts, 100 mg/L myo-inositol, 1 mg/L thiamine, 0.2 mg/L 2,4-dichlorophenoxyacetic acid, and 200 mg/L KH_2_PO_4_ (Sigma-Aldrich Inc., St. Louis, MO, USA) was used for wild type cells. BY-2 cells transformed with *XVE-NtPIN3b*-GFP [30] were cultured in MS medium supplemented with 40 µg/mL hygromycin and 100 µg/mL cefotaxime. To induce *Nt*PIN3b-GFP expression, MS medium was supplemented with 1 µM β-estradiol (10 mM stock solution in DMSO, Sigma-Aldrich Inc., St. Louis, MO, USA) at the beginning of the subculture interval. An appropriate amount of DMSO was added to the non-induced cells.

### 2.2. Preparation of Coverslips, Protoplasts and PM Ghosts

Coverslips for objectives with high numerical aperture (10 mm Ø, thickness n. 1.5H, 170 μm ± 5 μm; Paul Marienfeld GmbH & Co. Lauda-Königshofen, Germany) were coated with poly-L-lysine (Sigma-Aldrich, St Louis, MO, USA) for 45 min at RT, dried for 1 h at 60 °C and marked with laboratory tape (square 5 × 5 mm, Figure 1A), allowing later correct positioning on the light microscope stage and A-ESEM specimen chamber. Coverslips were placed in a 24-multiwell plastic plate (TPP Techno Plastic Products AG, Trasadingen, Switzerland) for further manipulation.

Protoplasts were isolated from 2-day-old cells following the previously published protocol [31,32]. The cell wall was digested by incubating the cells for 4 h in 0.45 M mannitol containing 1% cellulase “Onozuka” R-10 (Yakuruto Honsha Co., Ltd., Tokyo, Japan) and 0.1% pectolyase Y-23 (Kyowa Chemical Products Co., Ltd., Osaka, Japan). The released protoplasts were overlaid onto the MS medium supplemented with 0.4 M sucrose and centrifuged at 200× *g* for 10 min with the brake set at 0 (Hettich EBA 12 centrifuge, Andreas Hettich GmbH & Co. KG, Tuttlingen, Germany). The floating protoplasts were collected, filtered through a 100 µm nylon mesh and re-suspended in buffer containing 10 mM PIPES (MP Biomedicals, Santa Ana, CA, USA), 100 mM KCl and 285 mM mannitol, pH 6.8. Protoplasts were allowed to settle for 3 min (Sigma-Aldrich, St Louis, MO, USA) at poly-L-lysine coated coverslips (Figure 1B) and incubated for 2 min in lysis buffer (7 mM PIPES, 2 mM EGTA, 10 mM MgCl_2_, 1% DMSO, 6 mM DTT, 300 M PMSF, pH 6.9). Membrane ghosts were generated by several quick flicks of the coverslip (Figure 1C), ensuring the formation of a large number of PM ghosts [32]. PM ghosts adhered to the coated coverslips were fixed for 1 h in a solution containing 4% paraformaldehyde (prepared from EM grade 32% paraformaldehyde aqueous solution; Electron Microscopy Sciences, Hatfield, PA, USA) and 0.05% glutaraldehyde (Merck KGaA, Darmstadt, Germany) in PBS buffer (8 g/L NaCl, 0.2 g/L KCl, 0.158 g/L KH_2_PO_4_, 2.31 g/L NaHPO_4_***12H_2_O) and washed 3 × 10 min in PBS.

### 2.3. Immunostaining of PM Ghosts

Coverslips were transferred from the multiwell plate to a moisture chamber with parafilm on the bottom and incubated overnight at 4 °C in a drop (20 µL of a primary rabbit polyclonal anti-GFP antibody (cat. no. AS152987, Agrisera AB, Vännäs, Sweden), diluted 1:1000 in blocking solution (0.5% bovine serum albumin in PBS). After washing three times in PBS, the coverslips were incubated for 45 min at room temperature with Alexa Fluor^®^ 546-FluoroNanogoldTM IgG goat anti-rabbit IgG secondary antibody (cat. no. 7403, Nanoprobes, Yaphank, NY, USA) diluted 1:200 in 0.5% BSA. Since the size of the gold particles on the secondary antibody is 1.4 nm, a gold-enhancement reaction was performed to increase their size. The GoldEnhanceTM EM plus mixture (cat. no. 2114, Nanoprobes, Yaphank, NY, USA) was prepared immediately before use as described in the manufacture and published protocols [33]. Briefly, equal volumes of the four components (solutions A, B, C and D) were prepared in a total volume of approximately 40 µL per coverslip. A total of 10 µL of component A (enhancer, green cap) was mixed with 10 µL of component B (activator, yellow cap). After 5 min, 10 µL of component C (initiator, magenta cap) and 10 µL of component D (buffer, white cap) were added. After mixing, the entire mixture was added to the PM ghost preparations. During a period of about 2 min, the enhancement was monitored with a conventional light microscope and the reaction was stopped by washing the coverslips three times with dH_2_O as soon as the color of the PM ghost changed. These preparations were stored in a moisture chamber until microscopy was performed.

### 2.4. TIRF Microscopy

TIRF microscopy was performed on the Zeiss ELYRA PS.1 imaging platform built on a fully motorized inverted microscope Axio Observer. Since the coverslips must be carefully removed after TIRFM and stored for subsequent A-ESEM, they cannot be placed on a conventional slide during TIRF imaging. To properly define the position and to eliminate possible xy drift of the sample during imaging with the oil immersion objective, the coverslip with adhered cells was placed in the homemade aluminum microscope stage slide holder insert (Figure 1E). The immunostained cells adhered to the coverslip were overlaid with dH_2_O, and an overview tile-scan bright field image of the entire slide was acquired using the Plan-Apochromat 10× objective (NA 0.45, WD 2 mm). About 15 PM ghosts with the flattest appearance were selected, and their xy coordinates were saved for subsequent TIRFM. Individual ghost images were manually highlighted and calibrated using the Zen drawing and scale bar insertion tools. TIRFM was performed at selected positions with the alpha Plan-Apochromat objective 100× Oil DIC M27 Elyra (NA 1.46, WD 0.11 mm, used with Immersol 518 F for 30 °C). Images of the PM ghosts were acquired in two channels. The bright-field channel contained an overview of individual PM ghosts preselected with a low-magnification objective. Alexa Fluor 546 was excited with a diode-pumped solid-state laser 561 nm (200 mW; tuned to 0.4%), and fluorescence was recorded with the PCO Edge 5.5 scientific complementary metal-oxide semiconductor (sCMOS) camera using the BM570-620+LP750 beam splitter (16-bit, pixel size 60 × 60 nm). After imaging, the coverslips were carefully removed from the holder insert using thin tweezers, and the immersion oil was removed using isopropanol. All coverslips were placed individually in the multiwell plate and dried for 48 h at RT without the addition of any organic solvent.

### 2.5. A-ESEM and CLEM

A-ESEM is the next generation of the ESEM developed for imaging samples beyond the capabilities of commercial instruments. It combines a number of technological and methodological improvements that have been developed in ISI ASCR. For A-ESEM observation, a self-modified Quanta™ 650 FEG microscope (Thermo Fisher Scientific, Waltham, MA, USA) was used. The microscope is equipped with a newly developed differentially pumped chamber in the objective to optimize/minimize gas flow in the optical axis of the microscope. This enables to increase the detected signal-to-noise ratio as well as to achieve higher spatial resolution. Samples were imaged using a patented (European Patent Number: 2195822) high-efficiency Ionization Secondary Electron Detector with an electrostatic Separator (ISEDS), which enables low-dose imaging with higher resolution [34]. Two self-modified CCD cameras were used for sample monitoring during initial pumping of the A-ESEM and pre-adjustment of the correct sample position for correlative workflow. For the A-ESEM, coverslips were placed on an aluminum stub and attached with double-sided carbon tape for SEM. PM ghosts were imaged at a beam energy of 10 keV, a beam current of 30 pA and a working distance of 8.5 mm. To prevent charging, the non-conductive samples were imaged at 200 Pa water vapor pressure in a microscope specimen chamber.

For the large field of view imaging, as well as for the fast and accurate localization of the previously selected PM ghosts and further correlation of A-ESEM and TIRF images, the software MAPS 2.5 (Thermo Fisher Scientific, Waltham, MA, USA) was used. First, the correct position of the coverslips in the specimen chamber was set, and the marked areas of the previously analyzed ghost images were found using the images from the CCD cameras in the A-ESEM. Then, the light microscope images taken previously by a light microscope with 10× and 100× objectives were uploaded and used to match the sample and locate the analyzed ghosts. Finally, the TIRFM images were uploaded and aligned to A-ESEM images. The exact superposition of all images was determined using the three-point alignment method based on translation, rotation and resizing in two directions. The superimposed A-ESEM and TIRF images were used to correlate the position of gold nanoparticles, of which the positions and identities were acquired with the ISEDS.

### 2.6. Image Analysis and Statistics

To quantitatively assess the distribution of NtPIN3b-GFP within the PM of tobacco cells, interactive image analysis of CLEM images was performed. Fluorescence intensity profiles across nanodomains were generated from the TIRFM images of PM ghost using the original Zen black software (Carl Zeiss AG, Jena, Germany), profile plots were analyzed using OriginPro (OriginLab Corporation, Northampton, MA, USA) and all plots and statistics were generated using Sigma Plot (Systat Software Inc, San Jose, CA, USA). The frequency distribution of Full Width Half Maxima (FWHM) values (n = 850 from 13 PM ghosts) and the number of gold particles per nanodomain (n = 250 from 5 CLEM PM ghosts) were fitted with non-linear regression using 3-parameter Gaussian fits (f = a*exp(−0.5*((x−x0)/b)^2)) after passing a Shapiro–Wilk nonparametric normality test. The dimensions of individual gold particles corresponding to individual PM fluorescence spots were manually quantified in the software NIS Elements (Laboratory Imaging, Prague, Czech Republic). An equivalent diameter parameter, i.e., the diameter of the circle with the same area as the measured objects, was used.

## 3. Results

### 3.1. Immunolocalization of GFP-Tagged Auxin Efflux Carrier NtPIN3

As frequently reported in various immunofluorescence studies, the identity and specificity of the fluorescence signal must be optimized before conclusions can be drawn. To immunostain *Nt*PIN3, the primary anti-GFP antibody was used to immunolocalize GFP, which serves in *Nt*PIN3b-GFP as a fluorescent tag inserted within the cytosolic loop of the protein. Before performing immunostaining, we have considered the molecular size of the anti-GFP primary antibody and Alexa Fluor^®^ 546-FluoroNanogoldTM IgG secondary antibody. Their molecular size appeared to be suitable for detecting integral PM protein with the estimated size 10 nm in diameter (based on the secondary amino acid structure and available 3D models). As described in detail in the Materials and Methods, gold enhancement was performed to increase the size of the gold nanoparticles. To determine the correct timing for gold enhancement, a series of preliminary A-ESEM imaging runs were performed and the timing was optimized so that we have achieved the diameter of the gold-enhanced nanogold particles to be on average 38 nm ± 1.6 nm (CI, 95%). Our protocol detected gold nanoparticles located at a distance from the studied protein, allowing the staining of individual *Nt*PIN3b-GFP molecules (Figure 1D).

Protoplasts were isolated from β-estradiol-induced exponentially growing cells after microscopic detection of the presence of *Nt*PIN3b-GFP at the PM. PM ghosts released from protoplasts were adhered to poly-L-lysine-coated coverslips and immunostained. These preparations allowed to perform TIRFM in well-adhered, thin layers of PM and PM-associated structures, without any interference of the cell wall structures. The observation by TIRFM was performed in a custom aluminum stage insert as described in the Materials and Methods. As shown in Figure 2A and Figure A1A, nanodomains of various sizes were observed in PM with FWHM ranging the most frequently between 300 and 500 nm (Figure 2B). Fluorescence structures observed by TIRFM represented molecules in close proximity to the coverslip and objective. Thanks to the very thin preparation of the PM ghost, there was no fluorescence detected in more than one TIRF layer. The patterned distribution of the signal was not observed in controls stained only with FluoroNanogold antibody (Figure A1B in Appendix A), in non-induced cells stained with both primary and secondary antibodies (Figure A1C) nor in immunostained wild-type tobacco BY-2 cells (Figure A1D). Based on all these controls, the immunostaining protocol was considered specific for staining the *Nt*PIN3-GFP auxin efflux carrier in the protoplast ghost PM, allowing subsequent observation by A-ESEM.

### 3.2. CLEM of NtPIN3b-GFP

A-ESEM imaging with ISEDS of dried samples allowed unambiguous identification of the gold nanoparticles coupled to the secondary antibody, as well as their size and distribution. As described in the Materials and Methods, software-assisted alignment of TIRF and A-ESEM images was performed using fiducials defined by overlaying light microscopy images and A-ESEM overview images (Figure 2C,D). CLEM images showed numerous gold nanoparticles spatially correlated with but also located outside the TIRFM-detected nanodomains (Figure 2E). To quantify the frequency distribution of the number of gold nanoparticles spatially correlated with the fluorescence of the nanodomains, the number and position of the particles were scored manually using an overlay image (Figure 2E). On average, every fluorescence spot corresponded to five nanogold particles (Figure 2F). However, the number of particles within the nanodomain varied between 2 and 23, reflecting a rather large heterogeneity in this parameter. Interestingly, there was never any fluorescence that correlated with individual gold nanoparticles. These results suggest that the fluorescence of nanodomains with a diameter of 200–600 nm corresponds to 2 to 20 or perhaps even more individual *Nt*PIN3b molecules.

## 4. Discussion

In our previous work, estradiol-inducible *Nt*PIN3b-GFP have been shown to be one of the main vegetative auxin efflux carriers in tobacco, being present in the PM of tobacco BY-2 cells and performing an auxin-transporting role [30]. The heterogeneity in the distribution of this protein within the PM of protoplasts, which we show here by TIRFM, is in good agreement with several recently published reports on the distribution of *Arabidopsis thaliana* PIN proteins within PM, including nanodomains of PIN3 in the hypocotyl epidermal cells shown by Airyscan confocal laser scanning microscopy [17] and clusters of PIN2-GFP in the root epidermal cells shown by transmission electron microscopy on immunostained platinum replicas [35]. Our results also indicate that nanodomain organization is maintained even in cells with the removed cell wall, which has been shown to be important for the dynamics and localization of proteins within PM, including PINs [17,35,36,37]. Recent results also suggested that the heterogeneity in the distribution of plant PM proteins is under control of phospholipids and associated protein kinases [38,39,40] and that this dynamic organization is linked to auxin signaling and PIN auxin efflux carriers [41,42,43]. Since our CLEM approach could be adopted for cells isolated, e.g., during single-cell transcriptomic efforts [44] in *Arabidopsis thaliana*, it might help in the deciphering of details of these signaling events that are restricted to numerous very small PM domains. Using the model system of tobacco cells, we now use this approach for studying the PM distribution of all tobacco PIN proteins.

The CLEM approach described here allowed us to analyze the distribution of gold nanoparticles representing individual *Nt*PIN3 molecules within the PM in the context of their previous TIRFM imaging. To our knowledge, this is the first effort to achieve a correlation between immunofluorescence and electron microscopy imaging of plant-integral PM proteins. As mentioned in the introduction, CLEM techniques are still very rare in plants and are usually technically demanding [23]. Thanks to the non-invasive character of A-ESEM, our approach does not need any additional sample processing after TIRFM imaging. In this way, the average number of auxin efflux carrier molecules within PM nanodomains could be determined quantitatively. Although the low-energy electrons detected in A-ESEM do not provide detailed structural information, the spatial resolution, which extends to the nanometer scale, and the unambiguous detection of gold nanoparticles make this approach very advantageous for the study of plant PM proteins. As shown in our TIRFM/A-ESEM CLEM images, gold nanoparticles were spatially correlated with fluorescence spots, but they were also present outside of these domains. This indicates that our TIRFM imaging was not sensitive enough to record weak signals coming from mostly individual molecules outside of the nanodomains or there could be some heterogeneity in the fluorochrome and nanogold conjugation of the secondary antibody [45]. It is possible that there might also be slight quenching of the fluorophore by gold particles, although our preliminary experiments showed that the TIRFM pattern is very similar when the secondary antibody was conjugated only with Alexa 488 or 546. Considering the accessibility of the epitope, for numerous studies, quantum dots fluorescent nanocrystals are used for CLEM [46]. However, if they are used as conjugates with antibodies, they do not represent any advantage in the labeling when considering epitope accessibility [47]. Moreover, one should also consider the fact that nanogold particles detected by A-ESEM that were not detected by TIRFM might originate from slightly more distant space from the PM, as A-ESEM has a very high depth of field. This might be improved upon in the future by detecting fluorescence by some 3D superresolution technique [48] that would allow visualizing even more distant areas. In this way, it will be possible to discriminate whether nanogold particles without fluorescence correlation are indeed bound within PM.

Compared to other CLEM methods utilizing high-energy electron microscopy (TEM/STEM) of chemically or cryo-fixed samples, post-fixed and coated with metals, our method does not require sample post-processing to obtain reliable information on the spatial distribution of gold nanoparticles. We tried to optimize our protocol for TIRF/STEM as well, but the technique of PM preparation was technically too demanding to be performed on formvar or carbon-coated grids. In principle, our TIRFM/A-ESEM CLEM might be further improved for the observation of fully hydrated or living samples. We now optimize the pumping process of the specimen chamber and the thermodynamic conditions close to the sample using self-developed hardware. Further optimization of the conditions will be carried out after analyzing results with the software TDS (Num Solution & ISI CAS, Czech Republic), which is based on ANSYS software simulations and online in situ measurement of the sample temperature and environmental humidity. Measurements will be performed via micro-sensors integrated into the self-developed Peltier cooling stage in the A-ESEM [49]. This all depends on the optimization of staining procedures for plant cells that would utilize in vivo labeling with antibodies or specific labeling with fluorescent ligands carrying nanogold or quantum dots [50]. The inspiration comes from breast cancer cells, where whole cells were visualized in a fully hydrated state with CLEM between fluorescence microscopy and ESEM [51].

## Figures and Tables

**Figure 1 biomolecules-11-01407-f001:**
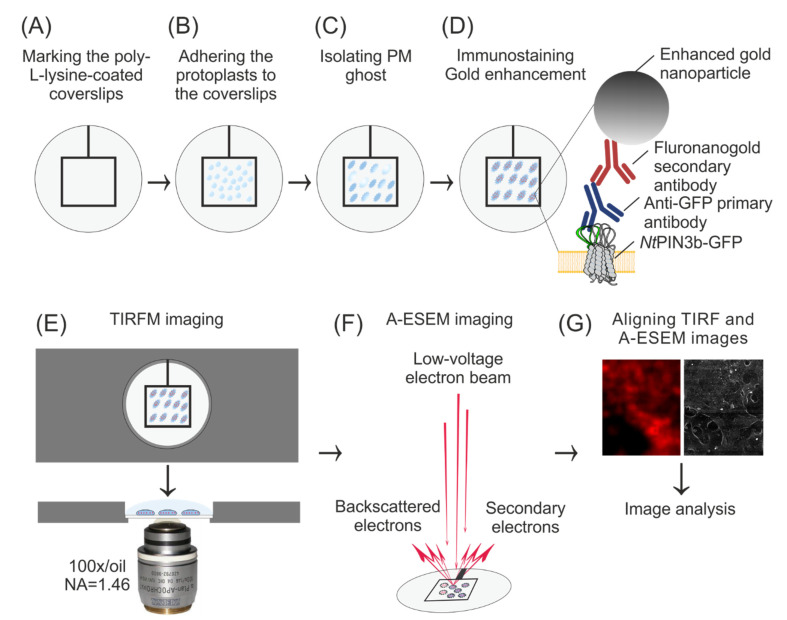
Workflow of TIRFM/A-ESEM CLEM of *Nt*PIN3b auxin efflux carriers in the PM of tobacco cell protoplast ghosts. (**A**–**C**) Poly-L-lysine coated coverslips are labeled with adhesive tape, protoplasts are adhered on coverslips and PM ghosts are isolated by several quick flicks. (**D**) Indirect immunofluorescence staining of *Nt*PIN3b-GFP. Protein, antibodies and gold after enhancement are shown in real size ratios. (**E**) TIRF microscopy on wet samples using a custom aluminum stage insert and a high numerical aperture objective. (**F**) A-ESEM performed on dried samples. (**G**) Software-assisted alignment of TIRFM and A-ESEM images using fiducials defined by superimposing bright field and A-ESEM overview images.

**Figure 2 biomolecules-11-01407-f002:**
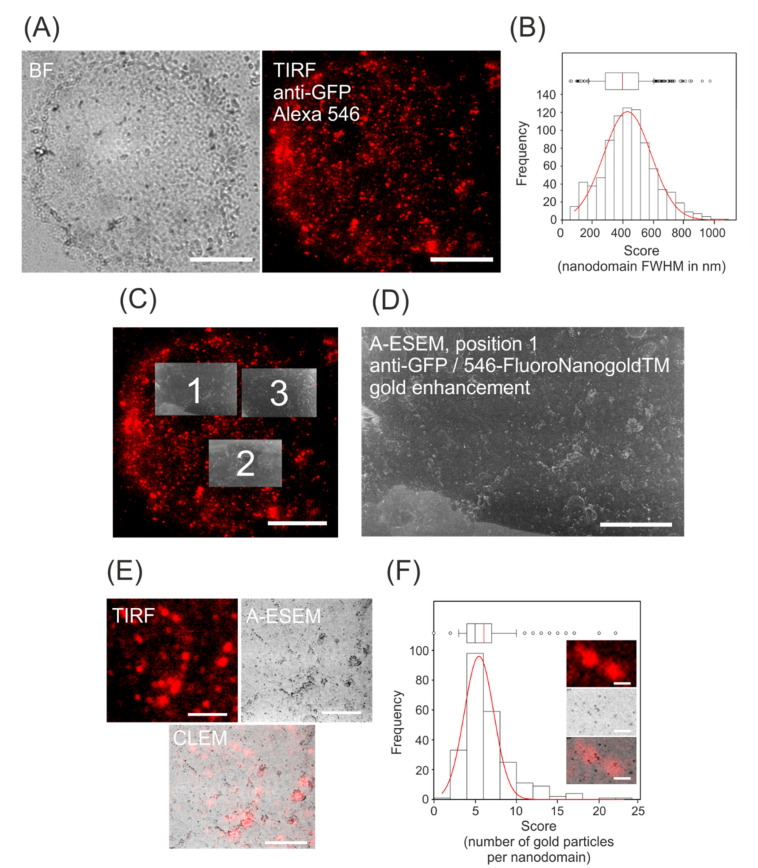
TIRFM/A-ESEM CLEM of auxin efflux carrier *Nt*PIN3b. PM ghosts from *Nt*PIN3b-GFP tobacco cells immunostained with primary anti-GFP antibody and secondary Alexa Fluor^®^ 546-FluoroNanogoldTM antibody. (**A**) PM ghost adhered to poly-L-lysine-coated coverslips imaged with alpha Plan-Apochromat objective 100x Oil DIC in bright field (**left**) and TIRF (**right**). Scale bar 10 µm. (**B**) Frequency distribution of full-width half maxima (FWHM) diameters of fluorescence spots representing PM nanodomains containing *Nt*PIN3b-GFP; n = 850, binned to 20 categories, 13 ghosts in total. (**C**) A-ESEM images superimposed on the TIRF image, three positions are marked 1-3. Scale bar 10 µm. (**D**) A-ESEM image of position 1; white dots represent enhanced nanogold particles. Scale bar 3 µm. (**E**) CLEM performed on the image of position 1. For better visibility during manual image analysis, the A-ESEM image is shown in complementary colors, while the gold particles are visible as black dots. Scale bar 1 µm. (**F**) Frequency distribution of the number of gold particles per nanodomain with *Nt*PIN3b-GFP; n = 250, binned to 12 categories, five individual CLEM images in total. Inset images show two brightly fluorescing spots on TIRF, containing about seven individual gold particles. Scale bar 500 nm. (**B**,**F**) Non-linear regression using three-parameter Gaussian fit is shown in red. The box plot indicates the 25th and 75th percentiles on its left and right boundaries, respectively. Black and red lines in the box mark median and mean values, respectively. Error bars indicate the 90th and 10th percentiles and individual points represent outliers.

## Data Availability

Data available in a publicly accessible repository, https://osf.io/bvs5d/.

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
