# Peer review of "Correlative Light-Environmental Scanning Electron Microscopy of Plasma Membrane Efflux Carriers of Plant Hormone Auxin"

_biomolecules, 2021, doi:10.3390/biom11101407_

Round 1

Reviewer 1 Report

This is an interesting paper and worthy of publication in the special issue of  ‘Biomolecules.’ I feel this will be a paper of interest to a reasonably large community of researchers.  On the whole, the paper is well written. I do have a few comments I’d like to see addressed before the manuscript is accepted. 
The correlated imaging method described in the paper is unique, and meets a need  in plant science.  There are few methods capable of imaging plant cells at high spatial resolution due to the physical characteristics of plant specimens.  In which case, the authors should scale back on their narrative that the alternative require, for example, ‘extensive fixation protocols’ (line 62, lines 326-328, and elsewhere).  Arguably, the specimens described in this work are also heavily processed and chemically fixed.  
The authors use phrases, such as ‘rather inhomogeneous distribution’  (line 12) and  ‘quite technically demanding’ (line 57).  Remove the modifiers ‘rather’ and ‘quite’.  It is either homogeneous or inhomogeneous, demanding or straightforward. 
Reword ‘plenty of biological roles’ (line 30) and give some concrete examples. 
Line 46, explain why correlation is difficult.
Line 216, describe the criteria by which the antibodies ‘appeared suitable’, and reword ‘around 10 nm’ to make it clear what you are referring to. 
Line 222, what is a ‘reasonable distance’? Criteria? 
Lines 242 - 256, I found this paragraph confusing to read.  It could be made significantly clearer by rewording.   Also, what is the confidence generating a ‘dried sample’ doesn’t significantly alter the dimensions and organization of the specimen.  
Line 311, wouldn’t it be expected there will be heterogeneity, rather than ‘not very probable’?  If not, why not?

I recommend publication in the special issue on 'Lateral Segregation of Molecular Components Enhances Functionality of Biological Membranes' after minor revision.

Author Response

Rev.1:

Comments and Suggestions for Authors

This is an interesting paper and worthy of publication in the special issue of  ‘Biomolecules.’ I feel this will be a paper of interest to a reasonably large community of researchers.  On the whole, the paper is well written. I do have a few comments I’d like to see addressed before the manuscript is accepted.

The correlated imaging method described in the paper is unique, and meets a need  in plant science.  There are few methods capable of imaging plant cells at high spatial resolution due to the physical characteristics of plant specimens.  In which case, the authors should scale back on their narrative that the alternative require, for example, ‘extensive fixation protocols’ (line 62, lines 326-328, and elsewhere).  Arguably, the specimens described in this work are also heavily processed and chemically fixed.

> Thank you very much for this kind evaluation and a valuable comment. Yes, we are aware of the fact that our samples are fixed and processed. We modified the text to better explain that our approach does not need any processing before electron microscopy.

The authors use phrases, such as ‘rather inhomogeneous distribution’  (line 12) and  ‘quite technically demanding’ (line 57).  Remove the modifiers ‘rather’ and ‘quite’.  It is either homogeneous or inhomogeneous, demanding or straightforward.

> We have removed these modifiers.

Reword ‘plenty of biological roles’ (line 30) and give some concrete examples.

> We have rewarded this expression stating three main “roles” of PM.

Line 46, explain why correlation is difficult.

> We have rephrased the sentence to show that it is mostly because of the lack of suitable CLEM protocols in plants.

Line 216, describe the criteria by which the antibodies ‘appeared suitable’, and reword ‘around 10 nm’ to make it clear what you are referring to.

> The criterion that is described here is purely described based on the size of antibody molecules, fluorescence tag and nanogold particles. We have rephrased the sentence to better explain this fact.

Line 222, what is a ‘reasonable distance’? Criteria?

> We have rephrased this sentence to better reflect the fact that we wanted to reach the situation that allows to stain individual NtPIN3b-GFP molecules.

Lines 242 - 256, I found this paragraph confusing to read.  It could be made significantly clearer by rewording.   Also, what is the confidence generating a ‘dried sample’ doesn’t significantly alter the dimensions and organization of the specimen.

> Thank you very much for this comment. Actually, when we first time generated a CLEM images, we were quite positively surprised by the fact that our RT drying of samples did not prevent unambiguous correlation to be achieved. Frankly, we did not expect this and were ready to work in hydrated samples, which however would be more difficult. So, the confidence is streaming from the fact that we achieved very robust correlation using always more fiducials. The quality of CLEM was usually very good for the majority of optical field, sometimes one could detect dimension distortions at the margins areas of the sample. However, this is not limiting the overall CLEM to be performed.

Line 311, wouldn’t it be expected there will be heterogeneity, rather than ‘not very probable’?  If not, why not?

> We have removed the statement „not very probable“. Although fluoronanogold conjugates are quite homogeneous with respect to the agreement between fluorescence signal and gold presence, we cannot exclude heterogeneity of course.

I recommend publication in the special issue on 'Lateral Segregation of Molecular Components Enhances Functionality of Biological Membranes' after minor revision.

> We thank rev. 1 for the constructive comments and hope that we have improved our manuscript to his/her satisfaction.

Reviewer 2 Report

The presented article is devoted to the development of high-resolution methods for the study of biological objects. The article is well structured, the material is logically presented, all materials and methods used in the work are described in detail. The article is of certain interest for the bio-medical community as an extension of the functionality for direct visualization and analysis of intermolecular interactions. Despite the good level of elaboration of the material, a number of points, in my opinion, require improvement.

  1. Line 17. The decoding of the abbreviation TIRPM is incomplete, the word "fluorescent" is missing.
  2. Please clarify Figure 2d: What is shown in the "enhanced gold" inset? Is this an image of a gold nanoparticle? Explain what the two gray circles in the window mean.
  3. Are there two images on Fig. 2g the same areas or they are just examples of obtained by different methods pictures?
  4. Please provide a clearer explanation of the AESEM method. There is no information available on the Internet on the well-established configuration of such devices. If you enter this type of equipment yourself, please describe in more detail what the technical modernization of a conventional ESEM consists of, or give a link to the manufacturer's website.
  5. On line 169, you mention a patented technique. Please specify the patent number (or give a link), or not mention patenting at all.
  6. The work mentions the software support of NIS Elements and gives a link to the laboratory. As far as I know, this product is developed by Nikon, Japan. Please clarify this issue, whether I am right and give the correct link.
  7. Line 209 gives the term "3.1. Immulocalization ...". Perhaps you mean Immunolocalization?
  8. Lines 251-253 - very poorly worded proposal, please reformulate.
  9. Please reformulate line 354: ...microscope that made the new AESEM possible. The word "possible", in my opinion, does not reflect the possibilities and prospects of applying the new technique.

Author Response

Rev. 2:

Open Review

Comments and Suggestions for Authors

The presented article is devoted to the development of high-resolution methods for the study of biological objects. The article is well structured, the material is logically presented, all materials and methods used in the work are described in detail. The article is of certain interest for the bio-medical community as an extension of the functionality for direct visualization and analysis of intermolecular interactions. Despite the good level of elaboration of the material, a number of points, in my opinion, require improvement.

> Thank you very much for this kind evaluation and valuable comments that as we hope make our manuscript better.

Line 17. The decoding of the abbreviation TIRPM is incomplete, the word "fluorescent" is missing.

> Thank you very much, we have corrected the TIRFM abbreviation description.

Please clarify Figure 2d: What is shown in the "enhanced gold" inset? Is this an image of a gold nanoparticle? Explain what the two gray circles in the window mean.

> Thank you very much, we understood that this comment goes to the schematic representation of our setup in Fig. 1d, not 2d. To improve the clarity, we have now removed any graphics from the inside of the gold nanoparticle and also renamed “enhanced gold” to “enhanced gold nanoparticle”.

Are there two images on Fig. 2g the same areas or they are just examples of obtained by different methods pictures?

> These two images are just examples, real CLEM is shown in Fig. 2e.

Please provide a clearer explanation of the AESEM method. There is no information available on the Internet on the well-established configuration of such devices. If you enter this type of equipment yourself, please describe in more detail what the technical modernization of a conventional ESEM consists of, or give a link to the manufacturer's website.

> Thank you very much for this comment. Yes, the instrumentation is developed in the ISI ASCR, based on the commercial ESEM microscope. We have now described this method in more details and included also some prospects on its further development into the discussion.

On line 169, you mention a patented technique. Please specify the patent number (or give a link), or not mention patenting at all.

> We have included the patent number.

The work mentions the software support of NIS Elements and gives a link to the laboratory. As far as I know, this product is developed by Nikon, Japan. Please clarify this issue, whether I am right and give the correct link.

> Yes, this software is produced by Czech company, as specified. This image analysis is now continuosly developed already from early nineties and has been attached to Nikon microscopes under the name NIS elements much later. It is still produced by Laboratory Imaging, Prague, Czech Republic.

Line 209 gives the term "3.1. Immulocalization ...". Perhaps you mean Immunolocalization?

> Thanks a bunch for this correction, the typo was not recognized during final polishing of the text. We have corrected it.

Lines 251-253 - very poorly worded proposal, please reformulate.

> We have reformulated this sentence.

Please reformulate line 354: ...microscope that made the new AESEM possible. The word "possible", in my opinion, does not reflect the possibilities and prospects of applying the new technique.

> We did not want to specify possibilities and prospects of AESEM, but wanted to acknowledge colleagues that made this all possible. The sentence is now modified.